# The Role of a Cytokinin Antagonist in the Progression of Clubroot Disease

**DOI:** 10.3390/biom13020299

**Published:** 2023-02-05

**Authors:** Jana Bíbová, Veronika Kábrtová, Veronika Večeřová, Zuzana Kučerová, Martin Hudeček, Lenka Plačková, Ondřej Novák, Miroslav Strnad, Ondřej Plíhal

**Affiliations:** 1Laboratory of Growth Regulators, Faculty of Science, Institute of Experimental Botany of the Czech Academy of Sciences, Palacký University, Šlechtitelů 27, CZ-78371 Olomouc, Czech Republic; 2Department of Biophysics, Faculty of Science, Palacký University, Šlechtitelů 27, CZ-78371 Olomouc, Czech Republic

**Keywords:** clubroot, *Plasmodiophora brassicae*, cytokinin, auxin, PI-55, photosynthesis

## Abstract

*Plasmodiophora brassicae* is an obligate biotrophic pathogen causing clubroot disease in cruciferous plants. Infected plant organs are subject to profound morphological changes, the roots form characteristic galls, and the leaves are chlorotic and abscise. The process of gall formation is governed by timely changes in the levels of endogenous plant hormones that occur throughout the entire life cycle of the clubroot pathogen. The homeostasis of two plant hormones, cytokinin and auxin, appears to be crucial for club development. To investigate the role of cytokinin and auxin in gall formation, we used metabolomic and transcriptomic profiling of *Arabidopsis thaliana* infected with clubroot, focusing on the late stages of the disease, where symptoms were more pronounced. Loss-of-function mutants of three cytokinin receptors, AHK2, AHK3, and CRE1/AHK4, were employed to further study the homeostasis of cytokinin in response to disease progression; *ahk* double mutants developed characteristic symptoms of the disease, albeit with varying intensity. The most susceptible to clubroot disease was the *ahk3 ahk4* double mutant, as revealed by measuring its photosynthetic performance. Quantification of phytohormone levels and pharmacological treatment with the cytokinin antagonist PI-55 showed significant changes in the levels of endogenous cytokinin and auxin, which was manifested by both enhanced and reduced development of disease symptoms in different genotypes.

## 1. Introduction

*Plasmodiophora brassicae* is an obligate biotrophic protist that is the causal agent of clubroot disease, one of the most devastating diseases in the Brassicaceae family. The clubroot disease is spread worldwide, probably as a result of the transport of diseased plant material by early colonists [1]. To date, considerable progress has been made in understanding the biology of this pathogen [2,3,4,5], but the complex mechanisms used by the pathogen during the infection process are still not fully understood [6]. It is becoming clear that the clubroot pathogen modulates host immunity responses, as suggested by the identification of a *P. brassicae* methyltransferase responsible for the methylation of salicylic acid (SA) and subsequent production of methyl salicylate (MeSA), leading to the control of free SA levels in infected galls [7].

As a biotrophic parasite, the clubroot pathogen requires a plant host to reproduce, and the infection process can be divided into two main phases: (I) the primary infection and (II) the secondary infection of the root cortex [8]. While the primary infection is restricted to the root hairs of the plant host, the secondary phase is primarily targeted to the cortex of hypocotyl and roots [9]. Infection by clubroot leads to profound changes in meristematic activity that imply changes in plant anatomy, and disease symptoms appear mostly at later stages, including increasing hypertrophy and hyperplasia of infected roots; the roots become swollen and form characteristic galls [10].

The clubroot pathogen and Brassicaceae crops establish a specific relationship where plant hormones are involved in triggering immune responses and orchestrating disease-phase-dependent transcriptomic changes [11]. In addition to SA, other groups of plant hormones are involved in different stages of disease progression, especially cytokinin (CK) and auxin, two key players in plant organ development [12,13]. By manipulating CK and auxin levels, the pathogen promotes hypertrophy and cell division by reprogramming meristematic activity in infected roots, leading to the formation of root galls [13,14,15]. Increased CK levels promoting cell division were observed in the early stage of clubroot disease in Arabidopsis. Conversely, reduced CK levels and concomitant downregulation in the expression of host plant CK sensing and metabolism-related genes were observed in the late stage of infection [12,13,16]. Shifts in CK and auxin homeostasis in developing galls can potentially influence the progression of the clubroot disease; it was reported by Siemens et al. that plants constitutively overexpressing cytokinin oxidase/dehydrogenase show reduced disease symptoms [13].

The plant hormone cytokinin plays a crucial role in various processes of plant growth and development [17]. In addition, in close cooperation with other plant hormones, CKs also play a role in plant defense, affecting many biotic and abiotic stress responses [18,19]. Their homeostasis is controlled by an interplay of multiple processes—the biosynthesis, irreversible degradation, formation, and breakdown of CK conjugates, and CK transport [20,21]. CK catabolism is primarily controlled by cytokinin oxidase/dehydrogenase (CKX; EC 1.5.99.12; [22]), a key enzyme in CK irreversible degradation. On the other hand, isoprenoid CK biosynthesis is catalyzed by adenosine phosphate isopentenyl transferase (IPT; EC 2.5.1.27; [23]), and this reaction represents the rate-limiting step in CK biosynthesis.

The perception of CKs in *Arabidopsis thaliana* is associated with three membrane-bound sensor histidine kinases (HKs), AHK2, AHK3, CRE1/AHK4, based on a two-component phosphorelay signal transduction system [24,25,26,27,28]. The signal perception in Arabidopsis is redundant; individual receptors contribute to different developmental and other processes, and double mutant combinations of *ahk* mutant alleles are viable, but show changes to root morphology or chlorophyll content [29]. AHK2/AHK3 combination in particular appears to play a dominant role in controlling root vs. shoot growth. The rosette of the *ahk2 ahk3* double mutant was reduced approximately to half the size of the wild type; this mutant line also formed fewer cells in the shoot and showed reduced chlorophyll content and faster primary root growth. While *CRE1/AHK4* is predominantly expressed in the root and can be found in the vascular cylinder, *AHK2* and *AHK3* show much broader expression patterns among different plant organs, especially in aerial parts.

Another approach to studying CK perception in plants is the use of CK agonists or antagonists, i.e., compounds that activate HK receptors or prevent agonist binding to the receptor, respectively, leading to the occupancy and blocking of the receptor’s active site. PI-55, 6-(2-hydroxy-3-methylbenzylamino)purine, is closely related to naturally occurring 6-benzylaminopurine (BAP), an aromatic CK commonly used in laboratory practice and plant micropropagation techniques. PI-55 is the first example of an anticytokinin compound that specifically interacts with the CRE1/AHK4 receptor and antagonizes the function of natural CKs [30]. In this study, PI-55 was used together with *ahk* double mutant lines to assess whether changes in CK homeostasis may affect the process of clubroot development.

## 2. Materials and Methods

### 2.1. Biological Material and Growth Conditions

The Columbia (Col-0) ecotype of *Arabidopsis thaliana* (L.) Heynh was used. *ahk2-2 ahk3-3*, *ahk2-5 cre1-2*, and *ahk3-7 cre1-2* mutant lines were kindly provided by Thomas Schmülling [29]. The studied plants were grown in a cultivation chamber on soil (mixture with perlite 1:1) at 22 °C under long-day conditions (16 h light/8 h dark). Clubroot-infected roots of the Chinese cabbage variety “Granaat” were kindly provided by Pavel Kopecký (Crop Research Institute, Olomouc, Czechia).

### 2.2. Preparation of the Cytokinin Antagonist PI-55

Synthesis of the PI-55 compound and its biological testing was performed as previously described [30]. Details of the chemical synthesis are provided in the Appendix A.

### 2.3. Inoculation of A. thaliana Plants and Pharmacological Treatment

The clubroot galls were stored at −20 °C. Resting spores were extracted by homogenizing mature clubroot galls in distilled water. The suspension was filtrated through gauze (25 μm pore width) and subsequently centrifuged three times (1000× *g*, 7 min). Then, 14-day-old Arabidopsis plants were inoculated with 0.5 mL of the prepared inoculum of *Plasmodiophora brassicae* (10^7^ spores mL^−1^). Subsequently, non-infected or infected Arabidopsis plants were treated with PI-55 three days after inoculation (dai) for the first time, or the control group was mock-treated with DMSO (0.1%). Treatments with the cytokinin antagonist (or the mock treatments) were performed 9 times over the period of 3 weeks by injecting the soil around each plant with 5 mL of either 1 or 10 µM solution of PI-55.

### 2.4. Quantification of Plant Hormones

Fresh plant material (roots + hypocotyls) was homogenized under liquid nitrogen and divided into CK and auxin parts with weights of approximately 10 mg per biological sample. CK samples were extracted in 1 mL of modified Bieleski buffer (methanol/water/formic acid, 15/4/ 1, *v*/*v*/*v*; [31]) using an internal standard of stable isotopically labeled CKs (0.4 pmol per sample of CK bases, ribosides, 9-glucoside, and 7-glucoside, and 1 pmol per sample of *O*-glucosides and nucleotides) to check the recovery of the purification step and to validate the determination. Samples were analyzed by ultraperformance liquid chromatography (Acquity UPLC^®^ I-class system; Waters, Milford, MA, USA) coupled to a triple quadrupole mass spectrometer equipped with an electrospray interface (Xevo TQ-S, Waters, Manchester, UK), via the method described previously [32]. Quantification was performed by multiple reaction monitoring of [M + H]^+^ and the appropriate product ion [32,33]; Masslynx software was used using a standard isotope dilution method. Auxin samples were extracted using 0.8 mL of 50 mM Na-phosphate buffer (pH 7.0; 4 °C) containing 0.1% diethyldithiocarbamic acid salt, using an internal standard of stable isotopically labeled auxins (5 pmol per sample; as previously described [34]). Samples were dissolved in 30 µL of 10% MeOH, injected onto a reversed-phase column (Kinetex 1.7 µm C18 100 Å, 50 × 2.1 mm; Phenomenex), and analyzed using an Acquity UPLC^®^ I-class system (Waters, Milford, MA, USA) coupled to XevoTM TQ-S (Waters, Manchester, UK) equipped with an electrospray interface (ESI), via the method described previously [34].

### 2.5. Quantitative Real-Time PCR

The Spectrum Plant Total RNA Kit (Merck, Darmstadt, Germany) was used for the isolation of total RNA from 100 mg (fresh weight) of the biological material (hypocotyls + leaves of 28 dai plants), according to the manufacturer’s instructions. To minimize contamination with genomic DNA, total RNA was treated with 1.5 μL TURBO DNase (Thermo Fisher Scientific, Waltham, MA, USA) with 1 μL of the enzyme for 30 min. Subsequently, 1.5 μg of extracted RNA was used for first-strand cDNA synthesis by RevertAid H Minus M-MuLV Reverse Transcriptase (Thermo Fisher Scientific, Waltham, MA, USA), according to the manufacturer’s instructions, and diluted to a final volume of 200 μL. qPCR analyses were performed in the StepOnePlus Real-Time PCR System (Applied Biosystem, Waltham, MA, USA). The reaction mixture consisted of 5 µL of gb SG PCR Master Mix (Generi Biotech, Hradec Králové, Czechia), 1 µL of gb Passive Reference Dye (Generi Biotech, Hradec Králové, Czechia), 0.6 µL forward primer (5 µM), 0.6 µL reverse primer (5 µM), and diluted cDNA (2 µL) in a final volume of 10 µL. PCR was performed according to standard procedures using the following amplification program: 95 °C for 10 min, followed by 40 cycles of 95 °C for 10 s, 65 °C for 15 s, and 72 °C for 25 s. Ct values were normalized to actin 7 and final values were calculated using 2^−∆∆Ct^ [35].

### 2.6. Measurement of Photosynthetic Activity

Control and clubroot-infected plants at 28 dai, 33 dai, and 41 dai were used for the maximum quantum yield of photosystem II photochemistry measurements (*F_V_/F_M_* = (*F_M_* − *F*_0_)/*F_M_*) using FluorCamFC 800 (PSI, Drásov, Czechia). Prior to the measurement, plants were dark-adapted for 20 min. The minimal fluorescence of the dark-adapted sample (*F*_0_) was determined by applying several µ-seconds-long measuring flashes (red light, 0.1 µmol photons m^−2^ s^−1^). The maximal fluorescence of the dark-adapted sample (*F_M_*) was measured using the 800 ms saturating pulse (white light, 2700 µmol photons m^−2^ s^−1^).

## 3. Results and Discussion

### 3.1. Phenotypic Characterization of Clubroot-Infected Cytokinin Receptor Mutant Lines

To better understand how modulations of cytokinin sensing and metabolism may contribute to the control of the proliferation of clubroot disease in the model organism *A. thaliana*, we took advantage of available *ahk* double mutant lines that exhibit homeostatic control of endogenous cytokinin levels that differ between genotypes [29]. In addition, we further modulated endogenous hormone levels by repeated application of a cytokinin antagonist, PI-55, during the infection phase of Arabidopsis growth. PI-55 represents a powerful tool for modulating cytokinin sensing, and its effects in controlling the root architecture through the manipulation of cytokinin status in affected plants have been demonstrated [30].

In this study, 14-day-old Arabidopsis wild-type plants and double mutant lines *ahk2 ahk3*, *ahk2 ahk4*, and *ahk3 ahk4* were inoculated with *P. brassicae* and disease progression was monitored at selected time points: 28, 33, or 41 days after inoculation (dai). In primary infection, the so-called root hair infection, there are visible subtle changes in the lower part of the root system. At 41 dai, we could observe prominent symptoms associated with later stages—the ground leaf rosette was deformed and the leaves were chlorotic, with already forming necrotic parts (Figure 1). The application of the cytokinin antagonist in the higher concentration used (10 µM) alleviated some of the symptoms of the disease. Both the wild type and the *ahk2 ahk4* mutant line appeared to be more vital, whereas the *ahk3 akh4* double mutant was less responsive to PI-55 treatment (Figure 1). To obtain more convincing evidence of the observed effects, we further measured the photosynthetic performance of control and clubroot-infected shoots, as described in the following section.

### 3.2. Photosynthetic Performance in Infected Plants and Effects of PI-55 Treatment

Infection with the clubroot disease in Col-0 and mutant genotypes progressively led to a decrease in photosystem II (PSII) function represented by a reduction in the maximum quantum yield of PSII photochemistry (*F_V_/F_M_*) in all Arabidopsis plants (Figure 2). Accordingly, the downregulation of photosynthetic genes, including genes of the Calvin cycle, chlorophyll biosynthesis, and genes of photosynthesis light reactions, was observed in the shoots of clubroot-infected plants in a transcriptomic study that also showed specific subsets of DEGs in the infected roots and shoots from 17 to 24 dai [3].

Representative images of *F_V_/F_M_* imaging in individual genotypes revealed that among cytokinin receptor double mutants, the *ahk2 ahk4* mutant plants showed the highest resistance to fungal pathogen proliferation. Similar to WT plants, in the *ahk2 ahk4* double mutant, we observed only mild disease symptoms in plants at 28 and 33 dai, indicating relatively slow disease progression under our experimental conditions (Figure 3). In stark contrast, in the case of *ahk3 ahk4* mutants, the impairment of PSII function was severe (Figure 2 and Figure 3). At 41 dai, photosynthetic activity in the *ahk3 ahk4* mutant plants expressed as *F_V_/F_M_* was below the limit of detection, and already at 28 dai, plants showed severe stunting, leaf curling, and chlorosis together with PSII impairment (Figure 3). In the *ahk2 ahk3* mutant plants, disease symptoms caused by *P. brassicae* infection were similar, but the decline in *F_V_/F_M_* was not as pronounced (Figure 2 and Figure 3).

Application of the cytokinin antagonist PI-55 at both concentrations (1 and 10 µmol L^−1^) alleviated the decrease in *F_V_/F_M_
*values in both WT and *ahk2 ahk3* double mutants (Figure 2B,C and Figure 3C), while there was no positive effect of PI-55 treatment in *ahk3 ahk4* double mutants, as *F_V_/F_M_* ratios were comparable to those of mock-treated (DMSO) plants (Figure 2B,C and Figure 3C).

Collectively, these results indicate that *P. brassicae* infection generates disease symptoms and affects PSII function and photosynthetic parameters primarily in plants where CRE1/AHK4 and AHK2 cytokinin receptors are present. This means that these receptors may represent primary targets that *P. brassicae* uses to modulate host responses via the cytokinin signaling pathway. It was previously reported by Siemens et al. that *CRE1/AHK4* is significantly upregulated in the early stage of infection, which was consistent with the increased level of the cytokinin response gene *ARR5* [13]. Interestingly, the CRE1/AHK4 and AHK2 receptors share a similar ligand binding spectrum and appear to be at least partially functionally redundant and significantly distinct from the AHK3 receptor. Compared with infected Col-0 plants, the *ahk3 ah4* double mutant line showed more pronounced disease symptoms and, importantly, did not respond to PI-55 treatment. Since PI-55 primarily targets the CRE1/AHK4 receptor [30], it can be assumed that the effect of the compound on the possible modulation of disease progression can be observed preferentially in WT and *ahk2 ahk3* double mutants treated with PI-55.

### 3.3. The Hormonal Content in Infected Galls and Effects of PI-55 Treatment

We have previously successfully developed a procedure for the highly reproducible detection of a complex pool of CK metabolites in milligram amounts of tissue (root + hypocotyl) in clubroot-infected plant material. As expected, *trans*-zeatin (*t*Z) and isopentenyladenine (iP) were confirmed as the most abundant active CKs in 28 dai plants, with *t*Z being slightly more abundant in both uninfected and infected tissues in all genotypes used in our study (Figure 4). This was in line with our previous observations, where these two CK free bases comprised the majority of the active CK forms, and the abundance of other free bases, *cis*-zeatin (*c*Z) and dihydrozeatin (DHZ), was ~100 fold lower [16]. The overall pattern in CK free base abundance at 28 dai was evidently decreasing compared to uninfected plants of the same age, corresponding to the strong downregulation of cytokinin homeostasis genes, as previously reported [13,16]. Similar trends were seen with other CK forms, although not with the same impact among the studied genotypes (Figure 4). While the concentrations of both CK ribosides and CK nucleotides only showed small changes after infection in Col-0 plants, all other genotypes, and *ahk2 ahk4* in particular, showed significant reductions. In contrast, the most abundant inactive CKs, CK *N*-glucosides, were reduced to a similar extent in all genotypes. Consistent with our previous observations, we could observe a small increase in CK *O*-glucoside conjugates, which was evident in the *ahk2 ahk3* double mutant (Figure 4).

All receptor double mutants showed significantly higher CK content (both before and after infection) compared to Col-0 plants, consistent with previous observations [29]. However, unlike the aforementioned study of Riefler et al., where younger Arabidopsis seedlings were used for CK quantification, we could observe highly elevated levels of both CK free bases, *t*Z and iP, primarily in the *ahk3 ahk4* double mutant (Figure 4). Similar trends were observed for CK precursors (ribosides and nucleotides), where CK levels in the *ahk3 ahk4* mutant plants were followed by the *ahk2 ahk3* genotype. Interestingly, the application of the cytokinin antagonist PI-55 led to different effects among the studied genotypes. Importantly, we observed significant changes in the content of CK free bases, *t*Z and iP—whereas, in clubroot-infected Col-0 and the *ahk2 ahk3* double mutant, the application of PI-55 led to a significant decrease in active CK levels, in the *ahk3 ahk4* mutant, the trend was opposite (Figure 4). In the case of the *ahk2 ahk4* mutant line, the concentrations of active bases were generally low in infected tissues compared to other mutant genotypes and similar to infected Col-0 plants. Notably, the *ahk3 ahk4* mutant showed a similar pattern to most other forms of CK—both 1 µmol L^−1^ and 10 µmol L^−1^ PI-55 treatment induced an increase in hormonal content in infected tissues compared to DMSO-treated plants.

Next, we evaluated the profiles of auxin, another plant hormone that may play an important role in gall formation (Figure 5). Auxin and cytokinin concentrations have previously been shown to change rapidly in developing root galls following *P. brassicae* infection [36]. Many genes of auxin homeostasis were upregulated in club development; it appears that the balance of auxins and cytokinins may play a regulatory role during disease proliferation [13]. The previous results suggested that free IAA levels do not change dramatically in infected tissues at later stages of infection [16]. Consistent with this, the IAA content of wild-type tissues was not significantly affected by clubroot infection (Figure 5). Similarly, *ahk* double mutants showed relatively little change, except for *ahk3 ahk4*, where approximately a 25% increase in IAA levels was observed. Treatment with PI-55 (10 µmol L^−1^) in infected tissues resulted in a relatively high, approximately 2.5-fold increase in auxin content in the *ahk2 ahk3* and *ahk2 ahk4* double mutant lines and approximately a two-fold increase in wild-type plants.

Taken together, our results indicate that the balance of cytokinin and, somewhat unexpectedly, also auxin, is affected in response to the cytokinin antagonist treatment and that these changes may be related to the cytokinin status of infected plants. In general, compared to Col-0 plants, our results show higher overall CK content, especially *t*Z and iP free bases, in all CK receptor double mutants. This was particularly evident in the *ahk3 ahk4* double mutant, where the free base content was relatively high regardless of whether PI-55 treatment was applied. Unexpectedly, photosynthetic parameters and plant fitness were relatively worse in this receptor mutant (Figure 2 and Figure 3), and PI-55 treatment could not rescue the photosynthetic performance. Conversely, the relatively low content of CK free bases (especially after PI-55 treatment) in the *ahk2 ahk4* double mutant correlated with better plant fitness and preservation of photosynthetic performance in treated shoots. This leads us to believe that the content of cytokinins, especially the most abundant active forms, *t*Z and iP, but also reflected in inactive CK forms (ribosides, nucleotides, and glucosides), is directly correlated with the progression of this disease, which further confirms the positive role of CK in the process. With the exception of the *ahk3 ahk4* double mutant, treatment with PI-55 mostly resulted in further reductions in free base content, although some variability was observed between the concentrations used. However, the effect of concentration did not seem to affect photosynthetic parameters, as both 1 and 10 µmol L^−1^ PI-55 treatments produced similar results (Figure 2). Unlike cytokinin, the exact function of auxin in later stages of infection is still not clear, but it has been hypothesized that elevated auxin may stimulate cell growth and promote cell wall extensibility, associated with cell expansion [37]. In our experiments, the levels of free IAA were only mildly affected in response to the infection process, but treatments with 10 µmol L^−1^ PI-55 seemed to produce more significant effects. This suggests that auxin homeostasis may also be affected by CK antagonist treatment, but further experiments will be needed to understand the hormone’s role in the process, as IAA levels change quite dynamically from early to later stages of infection [12].

### 3.4. Expression Analysis of Cytokinin-Related Genes in Clubroot-Infected Plants

In our experiments, the wild type and the *ahk2 ahk3* double mutant appeared to respond to PI-55 treatment, as reflected in the evaluation of their photosynthetic parameters (Figure 2 and Figure 3). While the treatment with PI-55 had no positive effect on disease progression in the *ahk3 ahk4* mutant line, in the case of *ahk2 ahk3* and Col-0 lines, 10 µM PI-55 treatment partially prevented chlorosis and increased the chlorophyll content in infected shoots (Figure 1). Furthermore, the wild type and the *ahk2 ahk4* double mutant appear to share similar phenotypes and photosynthetic parameters, as well as similar profiles of active CK forms. Therefore, to further evaluate the possible role of the cytokinin antagonist in the later stages of clubroot development, we used the wild type and the *ahk2 ahk3* double mutant line to determine the transcript levels of selected CK pathway-related genes previously associated with disease progression (Figure 6). At 28 dai, quantitative real-time PCR of infected and uninfected plant samples revealed that the general expression pattern clearly showed signs of overall downregulation of the cytokinin pathway. Although some *ARR* genes in wild-type infected plants were slightly upregulated, the changes in the expression levels of these genes were rather low and close to the detection limit (Figure 6 and Appendix A). Notably, the expression of *ARR5* was slightly upregulated in Col-0 infected plants, in agreement with previous transcriptomic analysis where higher expression was observed for *CRE1/AHK4*, *ARR5*, and *ARR10* [13]. By contrast, the *ahk2 ahk3* double mutant showed an approximately twofold reduction in *ARR5* expression levels in the infected tissues. Additionally, in the *ahk2 ahk3* double mutant, we could observe the downregulation of most cytokinin pathway genes, in agreement with previously published data [13,16]. PI-55 treatment in wild-type infected plants showed further downregulation of the expression levels of cytokinin oxidase/dehydrogenase gene *CKX1* and two cytokinin de novo biosynthesis genes, *IPT3* and *IPT5* (Figure 6). Interestingly, *CKX4* appeared to be relatively strongly upregulated by around fivefold in the wild type after PI-55 treatment; slight upregulation of *CKX4* at a later stage of infection was also reported in a transcriptomic study by Siemens et al. [13].

These results are in general agreement with previous studies that conclusively demonstrated that cytokinin homeostasis begins to be downregulated already at early stages of infection [13], and the expression of genes associated with cytokinin metabolism and signaling was strongly suppressed [16]. Differential analysis of genes in clubroot-infected resistant vs. susceptible lines of *Brassica rapa* L. and subsequent metabolomic analyses further explored the function of different classes of plant hormones in infection and highlighted their importance in triggering immune responses in the plant host [11]. Interestingly, regarding active CKs, the authors report high variability, especially in *t*Z levels during gall formation, which is consistent with *t*Z acropetal transport through the xylem [11]. It is therefore possible that an imbalance in *t*Z levels can significantly affect cambium activity and development.

Taken together, the relatively strong repression of genes related to cytokinin metabolism and sensing both in the wild type and the *ahk2 ahk3* double mutant line appears to be consistent with our metabolomic data, where we could observe a strong reduction in CK content after infection that was further promoted by PI-55 treatment. In particular, *IPT3* and *IPT5*, together with *IPT7*, are the major players in root growth and lateral root organogenesis [38,39]. Temporal regulation of isopentenyl transferase genes in clubs of *B. rapa* has been described, which is associated with disease development [40]. Importantly, although the quadruple *ipt1;3;5;7* mutant was able to develop some symptoms of clubroot disease, club formation was severely compromised [16]. Thus, it has been speculated that CKs may directly regulate plasmodial development, and reduced CK levels may interfere with the development of the vascular cambium, which is essential for gall formation [16].

## 4. Conclusions

Our data from *ahk* double mutants infected with *P. brassicae* and PI-55 treatments strongly support a role of the plant hormone cytokinin in the development of clubroot disease. Higher cytokinin content in infected roots, as observed in the *ahk3 ahk4* double mutant, coincided with better disease progression. PI-55 treatment led to changes in the internal pools of cytokinin and auxin, which further affected disease symptoms. Thus, it seems that cytokinin antagonist treatment may lead to a further decrease in endogenous CK content and a temporarily increased auxin status in infected plants, and the disrupted balance of these two phytohormones in developing clubs may modulate the growth of the biotrophic pathogen at later stages of infection. Our results demonstrate that changes in cytokinin perception through genetic or pharmacological approaches are a promising strategy to alter endogenous cytokinin (and possibly also auxin) levels, potentially allowing increased resistance to the infection and delayed progression of clubroot disease symptoms.

## Figures and Tables

**Figure 1 biomolecules-13-00299-f001:**
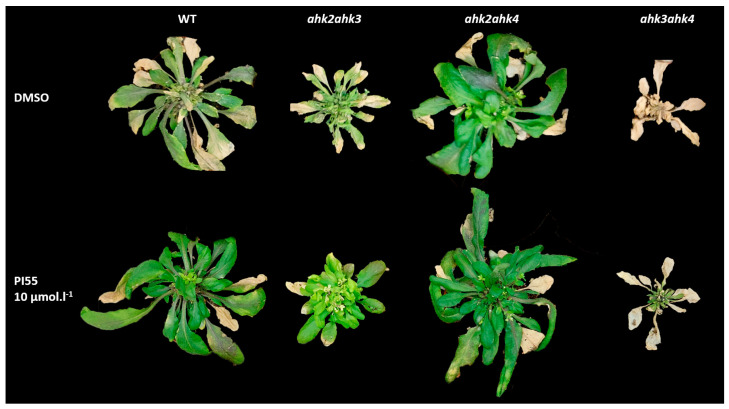
Comparison of clubroot disease symptoms in wild-type (Col-0) and *ahk2 ahk3*, *ahk2 ahk4*, and *ahk3 ahk4* double mutant lines. The 14-day-old plants were infected with *P. brassicae* and mock-treated with 0.1% DMSO or the cytokinin antagonist PI-55 at a concentration of 10 µmol L^−1^. Images of representative plants 41 days after inoculation (dai) are shown.

**Figure 2 biomolecules-13-00299-f002:**
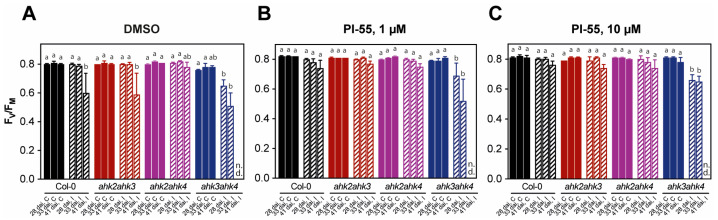
The maximum quantum yield of PSII photochemistry (*F_V_/F_M_*) in non-infected control wild-type (Col-0) or mutant Arabidopsis plants (C) and in respective plants infected with *Plasmodiophora brassicae* (I) 28, 33, and 41 days after infection (dai). Infected Col-0 and double mutant lines are indicated by diagonal hatching. Plants were mock-treated with (**A**) 0.1% DMSO or compound PI-55 at a concentration (**B**) 1 µmol L^−1^ or (**C**) 10 µmol L^−1^. Medians and quartiles are presented (n = 3–5). Different letters indicate a statistically significant difference (Tukey’s test; *p* < 0.05) between individual genotypes treated with DMSO or PI-55 (1 or 10 µmol L^−1^) within dai. n.d., not detected.

**Figure 3 biomolecules-13-00299-f003:**
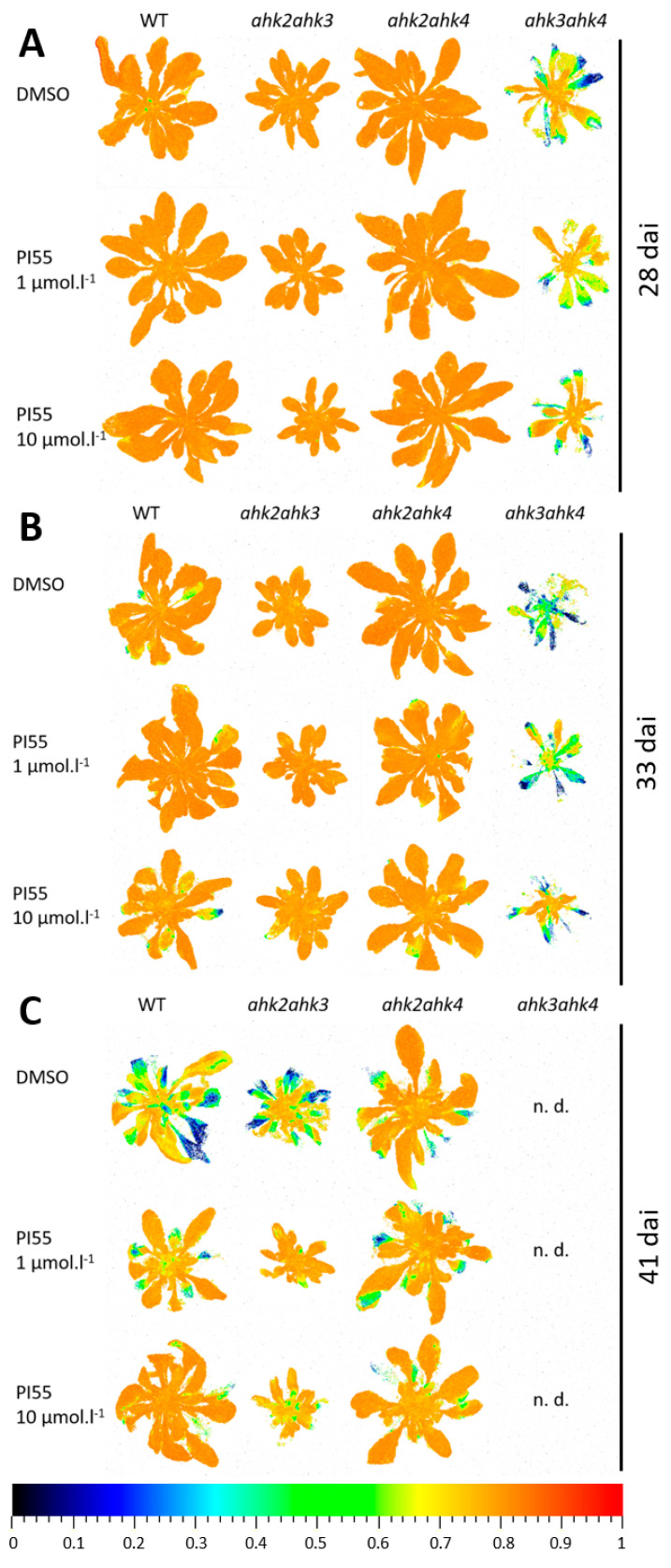
Imaging of maximum quantum yield of PSII photochemistry (*F_V_/F_M_*) in representative Arabidopsis wild-type (WT) or mutant plants at (**A**) 28, (**B**) 33, and (**C**) 41 days after infection (dai) with *Plasmodiophora brassicae*. Plants were mock-treated with 0.1% DMSO or the cytokinin antagonist PI-55 at a concentration of 1 or 10 µmol L^−1^. n.d., not detected.

**Figure 4 biomolecules-13-00299-f004:**
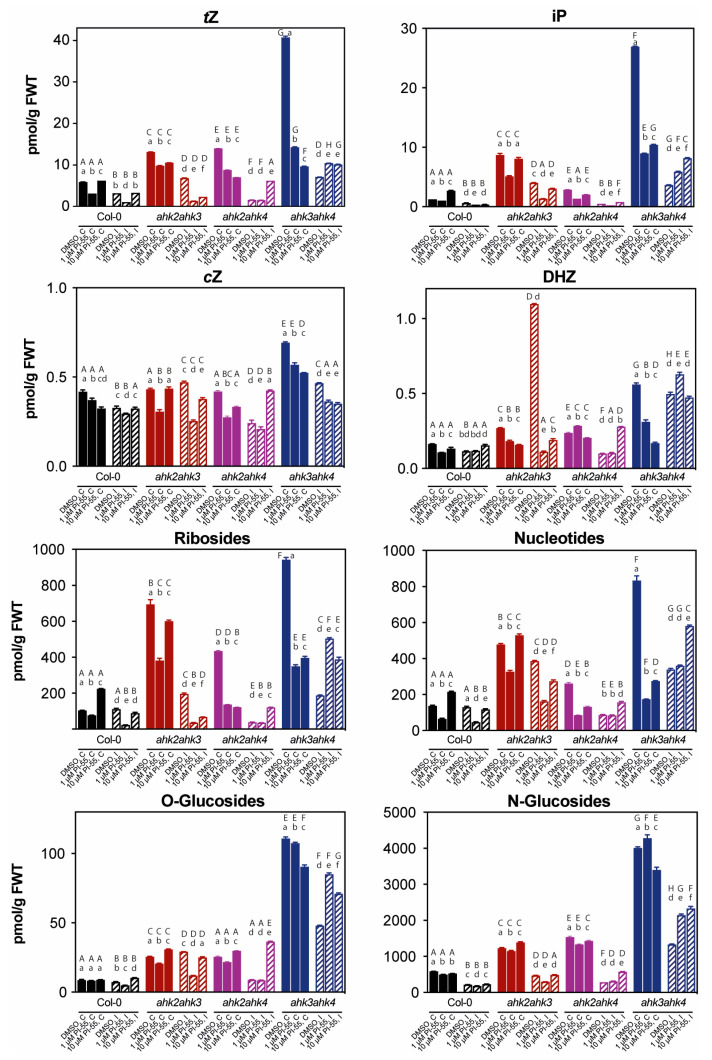
The cytokinin content of control and clubroot-infected wild type (Col-0) and the *ahk2 ahk3*, *ahk2 ahk4*, and *ahk3 ahk4* double mutant lines. Active (*t*Z, iP, *c*Z, and DHZ), precursor (ribosides and nucleotides), and conjugated (*O*-glucosides and *N*-glucosides) cytokinin content of control (C) and infected (I) root + hypocotyl tissue in 28 days after infection (dai) plants are shown. Plants were mock-treated with 0.1% DMSO or the cytokinin antagonist PI-55 at a concentration of 1 or 10 µmol L^−1^. Infected Col-0 and double mutant lines are indicated by diagonal hatching. Results are means (±SD) of 3 independent replicates expressed as pmol g^−1^ fresh weight. Different letters indicate a statistically significant difference (Tukey’s test; *p* < 0.05) within individual genotypes (lowercase letters) or between treatments with DMSO or 1 or 10 µmol L^−1^ PI-55 (uppercase letters).

**Figure 5 biomolecules-13-00299-f005:**
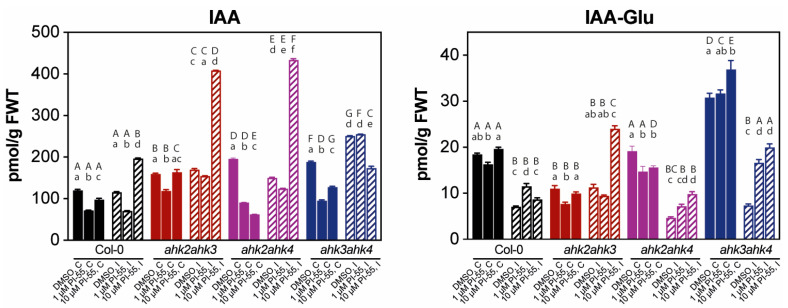
IAA and IAA-Glu profiles in wild type (Col-0) and the *ahk2 ahk3*, *ahk2 ahk4*, and *ahk3 ahk4* double mutant lines. Content of active and conjugated forms of IAA of control (C) and infected (I) root + hypocotyl tissue in 28 days after infection (dai) plants are shown. Plants were mock-treated with 0.1% DMSO or the cytokinin antagonist PI-55 at a concentration of 1 or 10 µmol L^−1^. Infected Col-0 and double mutant lines are indicated by diagonal hatching. Results are means (±SD) of 3 independent replicates expressed as pmol g^−1^ fresh weight. Different letters indicate a statistically significant difference (Tukey´s test; *p* < 0.05) within individual genotypes (lowercase letters) or between treatments with DMSO or 1 or 10 µmol L^−1^ PI-55 (uppercase letters).

**Figure 6 biomolecules-13-00299-f006:**
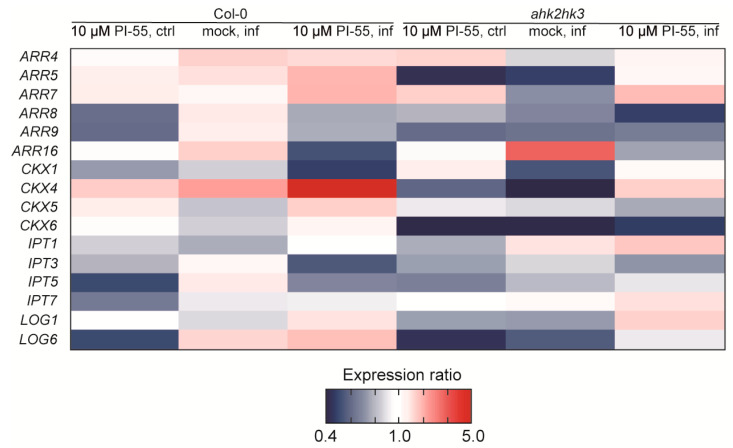
Expression profile of selected genes related to CK metabolism and sensing in the wild type (Col-0) and the *ahk2 ahk3* double mutant. The 28 days after infection (dai) control (ctrl) or clubroot-infected (inf) plants were mock-treated with 0.1% DMSO (mock) or the cytokinin antagonist PI-55 at a concentration of 10 µmol L^−1^. A heat map showing the relative expression of up- (red) and downregulated (blue) genes is color-coded according to the scheme shown in the figure. Expression changes are presented relative to either uninfected Col-0 or *ahk2 ahk3* plants that were mock-treated with 0.1% DMSO.

## Data Availability

The data that support the findings of this study are available from the corresponding authors upon reasonable request.

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
