# Peer review of "The Role of a Cytokinin Antagonist in the Progression of Clubroot Disease"

_biomolecules, 2023, doi:10.3390/biom13020299_

Round 1
Reviewer 1 Report
In this paper the authors investigate the role of cytokinin and auxin in gall formation resulting from clubroot infection. They use the model system Arabidopsis thaliana and applied metabolomic and transcriptomic profiling. As the homeostasis of two plant hormones, cytokinin and auxin, appears to be crucial for club development they work with loss-of-function mutants of three cytokinin receptors to study the homeostasis of cytokinin in response to disease progression. Quantification of phytohormone levels and pharmacological treatment with the cytokinin antagonist PI-55 showed significant changes in the levels of endogenous cytokinin and auxin, manifested by both enhanced and reduced development of disease symptoms in different genotypes. The authors conclude that “that changes in cytokinin perception through genetic or pharmacological approaches are a promising strategy to alter endogenous cytokinin (and possibly also auxin) levels, potentially allowing increased resistance to the infection and delayed progression of clubroot disease symptoms.
I find this a well designed study, scientifically sound with conclusive results, contributing significantly to the understanding of the role of cytokinin in gall formation and hinting to potential avenues for plant treatment.
I have a few minor remarks/suggestions that the authors might want to consider:
In “2.2. Preparation of the Cytokinin Antagonist PI-55” only a short reference to [30] is given. While I understand that the full synthesis of the PI-55 compound and its biological testing is described in the cited paper, I would ask the authors to consider to add a brief summery in addition to the reference, as in the cited paper the synthesis is only given in the supplement and the paper contains a couple of assays, some of them referring again to even older papers, not given the details (Standard bioassays based on stimulation of cytokinin-dependent tobacco callus growth, the retention of chlorophyll in excised wheat leaves and the dark induction of betacyanin synthesis in Amaranthus cotyledons were carried out as described previously). While the original work should be of course cited, a paper should also make it easy to repeat the experiments and not force to trace back what has been actually done. Also, if indeed the assays used would be the ones described in a earlier paper, the earlier paper should be cited for those. There should also be a clear statement if there were any deviations from the previous work or none.
There are two paragraphs numbered 2.2 (2.2. Preparation of the Cytokinin Antagonist PI-55 95 AND 2.2. Inoculation of A. thaliana plants and Pharmacological Treatment), the second one should be 2.3 and the following ones should be renumbered
In Results and Discussion the authors state: “Thus, it is tempting to speculate that further reduction in CK content and temporary elevated auxin status in PI-55-treated plants may shift the balance of cytokinins to auxins in developing clubs and modulate the growth of the biotrophic pathogen at later stages of infection.” If I understand correctly, this is new and hasn’t been proposed by other yet but the authors speculate so for the first time, correct? If so, I suggest to consider moving this statement to conclusions.
The authors also state at a later point that “Thus, it has been speculated that CKs may directly regulate plasmodial development and reduced CK lev-348 els may interfere with the development of the vascular cambium, which is essential for 349 gall formation.” In this case I believe the statement has been made by others in the work cited in the paragraph above the statement. A citation should be added where this statement comes from, or it should be made clear that the authors speculate based on the cited work.
Reviewer 2 Report
This manuscript “The Role of a Cytokinin Antagonist in the Progression of
Clubroot Disease” by Bibova and co-authors described the role of plant hormones in the development of clubroot disease. They have investigated the role of cytokinin and auxin in
gall formation using metabolomic and transcriptomic profiling in Arabidopsis thaliana. They have infected Arabidopsis plants with clubroot disease to find the role of CK and Auxin. I really appreciate the author and co-author for writing a well-organized manuscript.
Generally (in the case of Ck and auxin), it is perceived that both hormones directly and negatively regulate the signaling of the other. But they have investigated that the balance of both (cytokinin and auxin), is affected in response to the “cytokinin antagonist treatment”.
You know that plants often go through more than one biotic and abiotic stress, and it becomes difficult for a plant to deal with multiple stresses simultaneously. Plants use multiple signaling cascades including the interplay of multiple genes, biochemical and physiological processes set in motion for broad-spectrum resistance against the stresses. Each stress requires an exclusive acclimation response, tailored to the specific needs of the plant, but the combination of two or more stresses might require unique defense-responsive elements.
If possible, it’s just a suggestion. Can you find or at least predict with Insilco analysis (Using transcriptome or metabolome data) any clue or modify the CK-Auxin crosstalk pathway in response to cytokinin antagonist treatment”.
Its just a suggestion, not necessary.
Reviewer 3 Report
In this study, different cytokinin receptor-deficient genotypes and a cytokinin antagonist (PI-55) are used to analyze the function of cytokinins (CK) in clubroot disease caused by Plasmodiophora brassicae. The background of the study is clear and the results of the analysis are reliable. However, there is insufficient discussion regarding the relevance of differences in disease symptoms, endogenous levels of CK (and IAA), and expressions of CK-related genes exhibited by double mutants of CK receptors and their respective effects by the CK antagonist. For example, why does ahk3/ahk4 show severe symptom when CK signaling is suppressed, whereas the CK antagonist restores disease when AHK4 is functional in other mutants or wild type plant? Aren’t they the opposite results considering from the CK signalling? Discussion of the reasons for this inconsistent result seems to have been avoided.
Specific comments, including minor points, are listed below.
L89-90:
Why are different alleles used for AHK2 and AHK3 in the double mutants? (ahk2-2 vs ahk2-5, ahk3-3 vs ahk3-7). It would also be helpful for the readers to give information on the differences in the function of the three receptors, including phenotypes in each deficient mutant, and temporal and spatial expression profiles of each receptor gene.
L110-:
After homogenizing the analytical samples with liquid nitrogen, they are separated for IAA and CK analyses, but it is not likely that the samples can be weighed accurately in the frozen state. I wonder why internal standards of both CK and IAA were not added to the sample before homogenization.
Fig. 1 & Fig. 2:
From the photos, it is not possible to distinguish the difference between the mock-treated plants and those treated with PI-55. Also, the effect of infections is not significant in WT or ahk2ahk3 in Fig. 2 (both are marked with “a”). Is it OK to use this result as a basis for the discussion that follows (e.g., L323-324)? In L224, it is also described that "ahk2 ahk3 and ahk3 ah4 showed more pronounced disease symptoms," but ahk2/ahk3 does not seem to differ from WT.
Fig. 4:
- Line 245 states "overall pattern of decline" of CK levels, but it does not say that the decline is caused by infection.
- Student's t-test cannot be used as a statistical treatment in the results here. It is also inappropriate to show significant differences only for the data sets used in the discussion.
- Why are the amounts of riboside, ribotide, O-glucoside, and N-glucoside shown as the sum of CKs with and without hydroxyl groups in the side chains (no-hydroxy, trans-hydroxy, or cis-hydroxy)? There seems to be no reason why they should not be shown separately.
Fig. 6:
- In L330-331, it is described “Although some ARR genes in wild-type infected plants were slightly upregulated, the expression levels of these genes were rather low and close to the detection limit.” But information on the expression levels of each gene is not available from Fig. 6. The results should be included (at least as a supplemental data) to show the difference in relative expression levels of each gene.
- For each gene being analyzed, the function of each and how its expression level is affected by CK signaling should be described, rather than simply citing previous papers. For gene families, if tissue specificity of expression or differences in biological function are known, they should also be described. Based on such information, the expression levels of each gene should be discussed (e.g., the result that only CKX4 is upregulated by PI-55 among CKXs).
Reviewer 4 Report
The reviewed article presents important and interesting results of sophisticated experiments competently conducted by the authors. The article will be of interest to the readers, but still I advise authors to work with it a bit more, since some passages are not written clearly enough.
1. Line 80. I recommend either to specify that “agonists” mean compounds that can bind to and cause activation of a receptor, thus mimicking an endogenous ligand or delete this term (it is not mentioned in the article below).
2. Lines 173-174. “the leaves were chlorotic with already forming necrotic parts (Figure 1).” – I failed to see this in the figure. I advise either to provide photos of separate leaves with distinct signs of chlorosis and necrosis or to delete the statement (data on pigment content would be useful).
3. Just below “Both the wild type and the ahk2 ahk4 mutant line appeared to be more vital” – again this statement is not supported by the figure. The data on Fig. 2 and 3 are supporting this notion. If no clearer photos of whole plants are available, I think Figure 2 and 3 may be sufficient to support the statements. But reference to figure 1 is not convincing.
4. Section “3.2. Photosynthetic Performance…” starts with literature data. I recommend the authors to describe first their own data and then to support them with the references to the literature (the same concerns passages below.
5. Fv/Fm data demonstrate that “mutants, the ahk2 ahk4 mutant plants showed the highest resistance to fungal pathogen proliferation”. I recommend authors to relate the data on hormone assay to the differences in the resistance of distinct mutant to the infection. The same approach should be applied to the data on the effects of PI-55 on hormone content and plants resistance. Such comparisons are made in some cases, but not in all of them. It seems to me that it is important to emphasize the discussion concerning cytokinin content in high resistance ahk2 ahk4 mutant and its response to the action of PI-55.
6. Lines 221-223. “P. brassicae infection generates disease symptoms and affects PSII function and photosynthetic parameters primarily in plants where CRE1/AHK4 and AHK2 cytokinin receptors are present” - is this statement supported by the data showing higher resistance of ahk2 ahk4 missing both receptors? If this is so, I think this should be clearer stated.
7. Just the next sentence “The double mutant lines ahk2 ahk3 and ahk3 ah4 showed more pronounced disease symptoms and” – It is unclear what are these two mutants compared to?
8. It should be specified what the hatching of columns of the figures mean. Does this mean infected option? I failed to find any explanation.
9. I strongly recommend using ANOVA with Tukey’s test instead of t-test in the case of hormonal data (figures 4 and 5) as was done for Fv/Fm, (Fig. 2). This approach allows comparison of all means with each other. Otherwise comparison between genotypes remains not supported statistically (e.g., the statement that “All receptor double mutants showed significantly higher CK content (both before 254 and after infection) compared to Col-0 plants” – the word “significantly” implies that the difference has been statistically proven)
Line 244. “we could observe an overall decreasing pattern in the amount of CK free bases at 28 dai” – again it unclear which variants are compared supporting the statement about “decrease” in CKs. Is the effect of infection meant? This should be clarified
Line 265. “A similar situation was observed in the ahk2 ahk4 mutant line” – it should be clarified similarity to which variant is meant.
Line 322-323. “In our experiments, the wild-type and the ahk2 ahk3 double mutant appeared to respond to PI-55 treatment “ – I think that it should be specified how they responded. It seems to me that the treatment increased stability of photosynthesis in infected plants of these genotypes. If this is so, I think it should be stated clearer.
Presentation of PCR results should be clarified. I failed to find the data on non-infected plants untreated with PI-55. Apparently, the rest of the data were expressed as a ratio to this variant (calculated separately for each genotype), but if this is the case, then this should be explained in the caption to the figure.
Expression data were obtained for only one of the mutants. Its choice should be justified
Round 2
Reviewer 3 Report
The revised manuscript adequately addressed my comments.
I thank the authors for their appropriate responses.